# Normally-off Hydrogen-Terminated Diamond Field-Effect Transistor with SnO_x_ Dielectric Layer Formed by Thermal Oxidation of Sn

**DOI:** 10.3390/ma15145082

**Published:** 2022-07-21

**Authors:** Shi He, Yanfeng Wang, Genqiang Chen, Juan Wang, Qi Li, Qianwen Zhang, Ruozheng Wang, Minghui Zhang, Wei Wang, Hongxing Wang

**Affiliations:** 1Key Laboratory of Physical Electronics and Devices, Ministry of Education, School of Electronic Science and Engineering, Xi’an Jiaotong University, Xi’an 710049, China; thekeith@stu.xjtu.edu.cn (S.H.); yanfengwang@xjtu.edu.cn (Y.W.); genqiangchen@stu.xjtu.edu.cn (G.C.); wj17032021@stu.xjtu.edu.cn (J.W.); lq15829918579@stu.xjtu.edu.cn (Q.L.); zqw19971018@stu.xjtu.edu.cn (Q.Z.); wangrz@xjtu.edu.cn (R.W.); zhangminghuicc@mail.xjtu.edu.cn (M.Z.); wei_wang2014@xjtu.edu.cn (W.W.); 2Institute of Wide Band Gap Semiconductors, School of Electronics and Information Engineering, Xi’an Jiaotong University, Xi’an 710049, China

**Keywords:** thermal oxidation, SnO_X_, diamond, FET, normally-off

## Abstract

SnO_x_ films were deposited on a hydrogen-terminated diamond by thermal oxidation of Sn. The X-ray photoelectron spectroscopy result implies partial oxidation of Sn film on the diamond surface. The leakage current and capacitance–voltage properties of Al/SnO_x_/H-diamond metal-oxide-semiconductor diodes were investigated. The maximum leakage current density value at −8.0 V is 1.6 × 10^−4^ A/cm^2^, and the maximum capacitance value is measured to be 0.207 μF/cm^2^. According to the C–V results, trapped charge density and fixed charge density are determined to be 2.39 × 10^12^ and 4.5 × 10^11^ cm^−2^, respectively. Finally, an enhancement-mode H-diamond field effect transistor was obtained with a V_TH_ of −0.5 V. Its I_DMAX_ is −21.9 mA/mm when V_GS_ is −5, V_DS_ is −10 V. The effective mobility and transconductance are 92.5 cm^2^V^−1^ s^−1^ and 5.6 mS/mm, respectively. We suspect that the normally-off characteristic is caused by unoxidized Sn, whose outermost electron could deplete the hole in the channel.

## 1. Introduction

Compared with traditional semiconductor materials (e.g., Si, GaN, and SiC), diamond has many outstanding properties, including a large band gap (5.45 eV), high carrier mobilities (electron: 4500 cm^2^V^−1^ s^−^^1^, hole: 3800 cm^2^V^−1^ s^−1^), high thermal conductivity (22 W K^−1^cm^−1^), and a high breakdown field (>10 MV cm^−1^) [1], making it a potential wide band gap semiconductor material for next-generation high-frequency and high-power electronic devices [2,3,4]. Devices based on diamond have been studied for a long time, such as metal insulator semiconductor field effect transistors (MISFETs) [5], microelectromechanical systems [6], UV photodetectors [7], Schottky diode [8], biosensors, and so on. Among these fields, the hydrogen-terminated (H-diamond) MISFET has attracted a great deal of attention because of its potential applications in radio frequency power amplifiers [3]. Importantly, the discovery of two-dimensional hole gas under the H-diamond surface overcomes the problem that doped atoms are difficult to ionize at room temperature. The conductivity of H-diamond comes from a surface transfer doping process [9] and the holes residing in an accumulation layer at the hydrogenated diamond surface [10]. Additionally, the surface charge density located nearby the interface is around 10^13^ cm^−2^ [1]. In this case, the H-diamond MISFET device usually shows a depletion mode. However, an enhancement mode (normally-off) device is required for the power device to realize the system safety and energy saving. 

Recently, several strategies have been applied to realize an enhancement mode device, such as partial C–O bonds in the channel area [11], structure engineering [12,13], and special insulators [14,15]. At the same time, atomic layer deposition (ALD), metal-organic chemical vapor deposition, and radio-frequency sputter deposition (SD) have been used to deposit the dielectric layer. All these methods have promoted the development of the normally-off H-diamond MISFET. However, most of these methods need a high vacuum or high temperature which may damage the 2DHG channel and cause degradation of the device performance. In order to avoid performance degradation, an ALD- Al_2_O_3_ layer (2–5 nm) is deposited as a buffer layer to protect the C–H bond from SD plasma [16,17], which would complicate the fabrication process. Therefore, it is necessary to find a simple method to prepare a dielectric layer.

Up to now, our group has realized normally-off H-diamond MOSFETs with low work function materials (LaB_6_) [18], metal/insulator/metal/semiconductor structures [19], solution-processed SnO_2_ [20], and oxidation of Al and Ti [21,22]. We discovered that oxidation of low work function metal is an effective and simple method to achieve the normal-off operation. In this work, a SnO_x_ film formed by thermal oxidation of Sn was utilized as a dielectric layer to fabricate H-diamond FET, whose electric properties have been investigated.

## 2. Method

A 3 × 3 × 0.5 mm^3^ IIb-type high-pressure high-temperature (HPHT) single crystal diamond (100) was used as a substrate. Figure 1a shows the fabrication process of H-diamond MOSFET. The substrate was first cleaned with a mixed acid solution of H_2_SO_4_:HNO_3_:HClO_4_ = 31.2:36:11.4 for 1 h at 250 °C and then cleaned with a mixed alkali (NH_4_OH:H_2_O_2_:H_2_O = 4:3:9) at 80 °C for 10 min to get rid of the non-diamond phase. In the epitaxial layer growth process, the substrate was treated with hydrogen plasma (H-plasma) in an MPCVD chamber for 20 min to clean off the surface contamination. The growth temperature, thickness, and chamber pressure for the H-diamond epitaxial layer were 850 °C, 300 nm, and 70 Torr, respectively. H_2_ and CH_4_ flow rates were 500 and 5 sccm. After epitaxial layer growth, the substrate was treated with H-plasma for another 10 min to generate an H–C bond. After that, the sample was exposed to air for 5 h.

Then, Au was deposited as source and drain electrodes by photolithography and electron beam evaporation (EB). The thickness and space of electrodes (L_SD_) were 100 nm and 20 μm, respectively. UV/ozone irradiation was used to isolate the devices, and the channel area was protected by a photoresist. After that, 5 nm Sn was deposited using an electron beam evaporator, then oxidized on a hot stage at 100 °C for 24 h to form SnO_x_ film in air without removing the photoresist. Finally, 120 nm Al gate was deposited directly on the SnO_x_ layer using a self-aligned process. The gate length (L_G_)was 8 μm. Figure 1b shows the optical photo of the fabricated device. Figure 1c is the cross-section schematic of an H-diamond FET. The quality of the diamond was characterized by X-ray diffraction (XRD) using four-bounce Ge (2 2 0)-monochromated Cu-kα and with a 10 nm slit on the detector arm. The X-ray photoelectron spectroscopy (XPS) technique was implemented to characterize the SnO_x_ film. The electric properties of SnO_x_-based MOSFET were measured by Agilent B1505 at room temperature.

## 3. Result and Discussion

As shown in Figure 2a, the XRD full width at half maximum (FWHM) of the substrate is 0.0105°, which demonstrates the high quality of the single crystal diamond. XPS results of the thermal oxidized film are shown in Figure 2b. We can find not only an oxide peak (Sn^4+^) at a binding energy of 486.4 eV [23] but also a metallic peak of Sn at a lower binding energy of 484.6 eV [24]. The presence of the two peaks implies that the Sn film was not completely oxidized. This may be caused by the formation of a dense film in the beginning that prevents further oxidation.

Figure 3a is the gate leakage current density (J-V) curve of the SnO_x_/H-diamond MOS structure. The *J* value increases with the increase in gate biases from 0 to −8.0 V, and the gate leakage current density is 1.5 × 10^−5^ A/cm^2^ at a gate bias of −5.0 V. The maximum gate current density at −8.0 V is 1.6 × 10^−4^ A/cm^2^. The capacitance–voltage (C–V) curve of the device was measured at 5 MHz. There were obvious depletion and accumulation regions in Figure 3b (black line). The capacitance (*C_OX_*) value is 0.207 μF/cm^2^. According to the reported method [25], the flat band voltage (*V_FB_*) of this device is determined to be −0.27 V from the C–V curve where the second derivative of *C_OX_* is 0. The C–V curve shifted to the right-hand side relative to the theoretical *V_FB_* (−0.62 V), which indicated the negative fixed charge (*Q_f_*) in the insulator (Figure 3c) [15]. The fixed charge density in SnO_x_ could be calculated by the following equations: [26]
(1)$$V_{FB}=\phi_{MS}-\frac{Q}{C_{OX}}$$
(2)$$\phi_{MS}=\frac{W_{S}-W_{M}}{q}$$
(3)$$Q_{f}=\frac{Q}{q}$$
where *ϕ_MS_* is the contact potential difference between the gate electrode (Al) and the semiconductor (H-diamond); *Q* is the total fixed charge; *C_OX_* is 0.207 μF/cm^2^; W_S_ and W_M_ are the work function of H-diamond (4.90 eV) [27], and Al (4.28 eV) [28]; *q* is the elementary charge (1.6 × 10^−19^ C). *Q_f_* was calculated to be 4.5 × 10^11^ cm^−2^, which is lower than that in Y_2_O_3_ (1.0 × 10^13^ cm^−2^) [15], LaB_6_ (1.47 × 10^12^ cm^−2^) [18], and TiO_2_/Al_2_O_3_ (9.8 × 10^12^ cm^−2^) [17]. The origin of the high fixed-charge density results probably from Sn that is not completely oxidized.

In order to investigate trapped charge density in SnO_x_ film, CV curves were measured from different directions. As shown in Figure 3c, the black and red lines represent gate bias sweeping from positive to negative and following an opposite direction, respectively. The hysteresis shift during the bias voltage sweeping is 1.85 V. The origin of the hysteresis in the SnO_x_ film was believed to be the incompletely oxidized Sn. The trapped charge density (*Q_t_*) could be obtained by Equation (4)
(4)$$Q_{t}=\frac{C_{OX}\Delta V_{FB}}{q}$$

The *Q_t_* was calculated to be 2.39 × 10^12^ cm^−2^. CV curves measured at 10 kHz, 100 kHz, 500 kHz, 1 MHz, and 5 MHz were shown in Figure 3d.

Next, the I-V characteristic of the MOSFET was investigated by an Agilent B1505A (Agilent Technologies, Santa Clara, CA, USA) power device analyzer. The gate length and gate width of this device are 8 μm and 100 μm, respectively. The threshold voltage (V_TH_) is extracted from the transfer characteristics (Figure 4a) at V_DS_ = −10 V, which is −0.50 V. The transconductance (g_m_) keeps increasing when V_GS_ shifts from V_TH_ to the negative direction, reaches the maximum value of about 5.6 mS/mm at V_GS_ of −4.2 V, and then decreases. Figure 4b shows the gate current (I_G_) and on/off ratio of the H-diamond FET, and the gate voltages (V_GS_) were measured from 2 to −8 V. I_G_ was measured when the drain voltage (V_DS_) was −10 V, which was 6.8 × 10^−12^ A at V_GS_ = −8 V. Such a low large leakage current indicates that a dense, insulating oxide film was formed by thermal oxidation. The small leakage current makes the thermal oxidation SnO_x_ film suitable for H-diamond MOSFET with low power consumption. The on/off ratio was evaluated to be ~10^8^, which is high enough for practical applications. In combination with the XPS results, the normally-off characteristic may be caused by partial oxidation of Sn.

Figure 5 shows the drain-source current density (I_D_) versus drain-source voltage (V_DS_) output characteristic curves at different gate voltage (V_GS_) varying from 2 to −5 with a step of −0.5V. With the increase in |V_GS_|, |I_D_| increased, indicating that it was a p-type channel. Additionally, the distinct pinch-off and saturation characteristics were observed in Figure 4a. The maximum drain current (I_Dmax_) is −21.9 mA/mm at V_GS_ = −5 V and V_DS_ = −10 V. Compared with the reported device in Table 1, this work has a relatively high current density and mobility. However, the current density is small, especially compared with a normally-on device (1.1 A/mm [29]). This is because, in normally-off devices, the current density is lower because the carriers in the channel are depleted by insulators or metal.

Equation (5) provided the relationship between effective mobility and *R_ON_* for the FET channel.
(5)$$R_{ON}=R_{CH}+R_{SD}=\frac{L_{G}}{W_{G}\times \mu_{eff}\times C_{OX}\times \left|V_{GS}-V_{TH}\right|}+R_{SD}$$
where *R_ON_* is on state resistance, *R_CH_* is the resistance of the channel, and *R_SD_* is the resistance between source/drain and H-diamond. *L_G_* is channel length, *W_G_* is channel width, and *C_OX_* is the capacitance of the MOS structure. Thus, the *μ_eff_* was calculated to be 92.5 cm^2^V^−1^ s^−1^, as shown in Figure 6. As a comparison, the reported *μ_eff_* of H-diamond MOSFETs are 38.7, 34.2, 39.0, and 88 cm^2^V^−1^ s^−1^ for the gate dielectrics of HfO_2_ [14], ZrO_2_/Al_2_O_3_ bilayer [30], SiN_x_/ZrO_2_ bilayer [31], and TiO_x_ [32], respectively. The high mobility suggests the thermal oxidized SnO_x_ film can form a good interface with the H-diamond surface, which provided a high electrical transport channel.

**Table 1 materials-15-05082-t001:** Summary of FETs properties with different gate insulators.

Ref.	Dielectric Layer	*L_G_* (μm)	V_TH_ (V)	C (μF/cm^2^)	I_DMAX_ (mA/mm)	*μ_eff_*(cm^2^V^−1^ s^−1^)
This work	SnO_x_	8	−0.50	0.21	−21.9	92.5
[20]	SP-SnO_2_	10	−0.50	0.46	−17.6	41.3
[18]	LaB_6_	2	−0.53	0.49	−57.9	195.4
[17]	TiO_2_/Al_2_O_3_	4	−0.8	0.83	−11.6	-
[33]	HfO_2_	5	−2.9	-	−11	38.7
[31]	SiN_X_/ZrO_2_	6	2.2	0.34	−28.5	39.0
[34]	MoO_3_	4	0.7	0.36	−33	108

## 4. Conclusions

In summary, we report herein an investigation of the H-diamond FET with SnO_x_ dielectric layer deposited by thermal oxidation of Sn. The SnO_x_/H-diamond MOS structure showed a low leakage current density, and the capacitance value is 0.207 μF/cm^−2^. A normally-off H-diamond FET was obtained with a V_TH_ of −0.5 V. Its I_DMAX_ is −21.9 mA/mm at V_GS_ = −5 V and V_DS_ = −10 V. The effective mobility and transconductance are 92.5 cm^2^V^−1^ s^−1^ and 5.6 mS/mm, respectively. According to the C–V results, the fixed charge density and trapped density are 4.5 × 10^11^ and 2.39 × 10^12^ cm^−2^, respectively. In combination with the XPS results, the normally-off characteristic may be caused by partial oxidation of Sn. This is a simple and effective strategy for the fabrication of normally-off H-diamond FET. However, it still has the problem of low current density, which may be improved by reducing contact resistance and scattering in subsequent studies.

## Figures and Tables

**Figure 1 materials-15-05082-f001:**
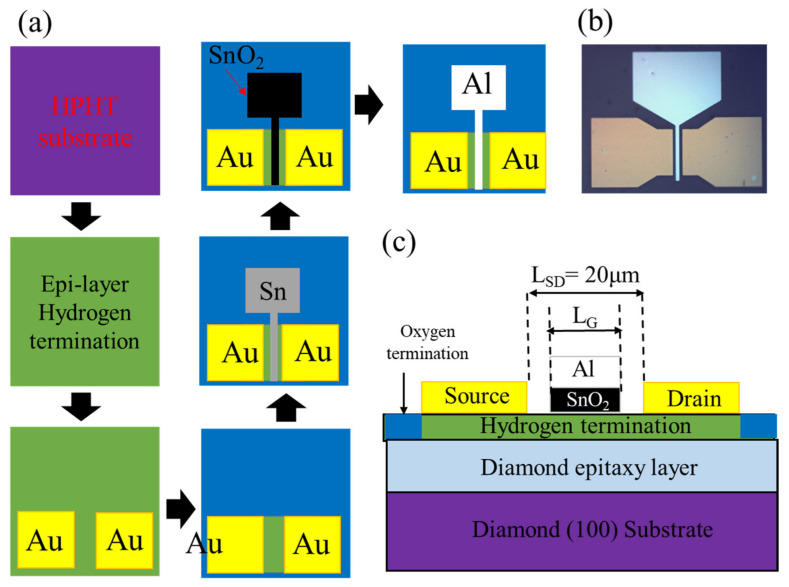
(**a**) Schematic of H-diamond FET fabrication process; (**b**) Optical photo of H-diamond FET. (**c**) Cross-section schematic of the device.

**Figure 2 materials-15-05082-f002:**
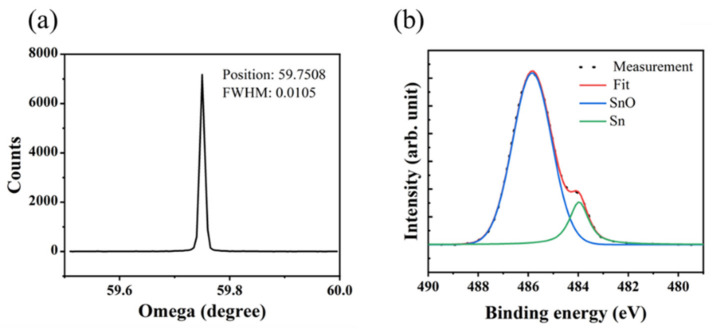
(**a**) XRD results of H-diamond, (**b**) Sn 3d photoelectron spectra for thermal oxidation of Sn.

**Figure 3 materials-15-05082-f003:**
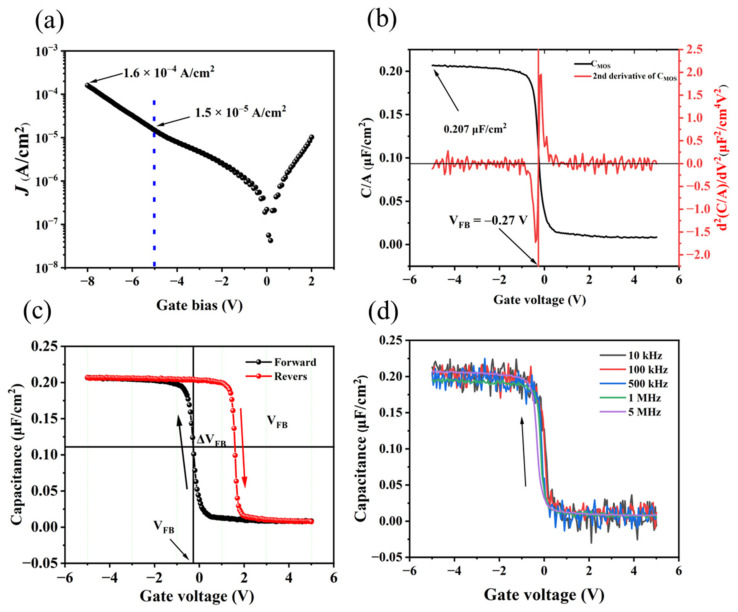
(**a**) J-V characteristic of SnO_x_/H-diamond MOS diode. (**b**) The solid line describes a capacitance voltage characteristic. The dotted line is the second derivative of capacitance as a function of gate voltage. (**c**) Capacitance voltage characteristics at 5 MHz (**d**) Capacitance voltage characteristics at different frequencies.

**Figure 4 materials-15-05082-f004:**
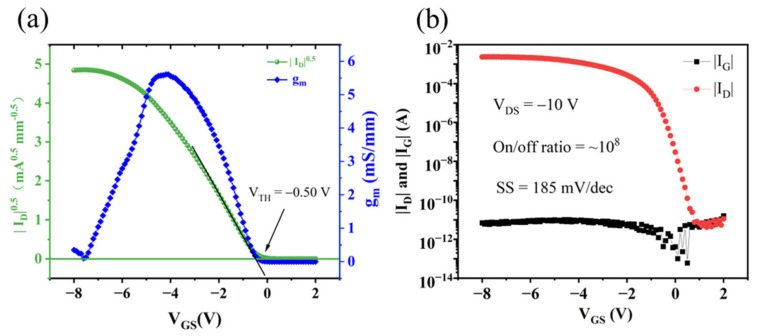
(**a**) transfer characteristics and transconductance of the SnO_x_/H-diamond based FET. (**b**) |I_D_| and |I_G_| in logarithmic coordinate.

**Figure 5 materials-15-05082-f005:**
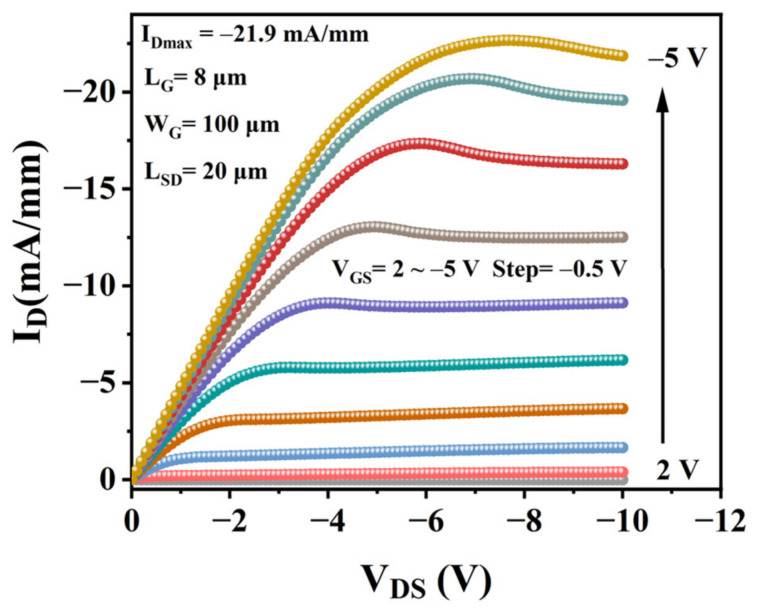
I_DS_-V_DS_ characteristic of SnO_x_/H-diamond based FET.

**Figure 6 materials-15-05082-f006:**
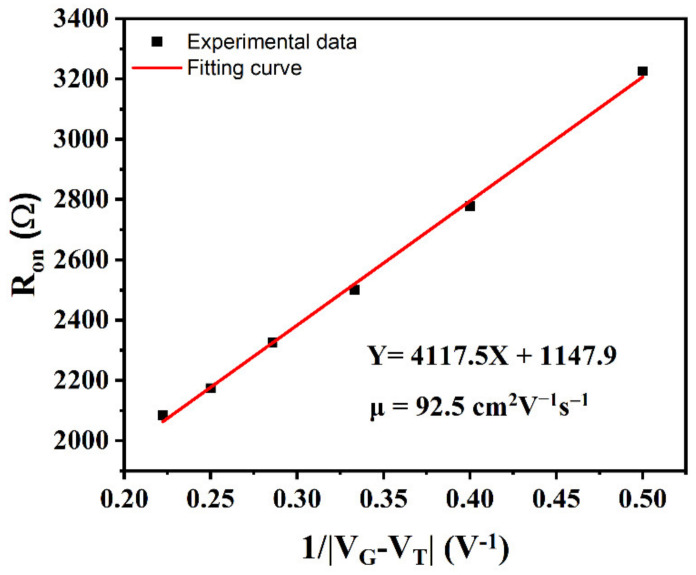
The plot of R_on_ versus 1/(V_G_−V_T_) and fitting curve.

## Data Availability

Data are contained within the article.

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
