# Peer review of "Normally-off Hydrogen-Terminated Diamond Field-Effect Transistor with SnOx Dielectric Layer Formed by Thermal Oxidation of Sn"

_materials, 2022, doi:10.3390/ma15145082_

Round 1
Reviewer 1 Report
This manuscript focuses on “Enhancement-mode hydrogen-terminated diamond field-effect 1 transistor with SnOx dielectric layer formed by thermal oxidation of Sn”. There are many similar works by the same Group. The following is my comments and questions.
1. Do you consider the topic original or relevant to the field? Does it address a specific gap in the field? This research is not very novel and similar with the previous reports by the same group
2. What does it improve the electrical properties compared with other published papers by your Group?
3. Could you give some comments or future plan to improve the electrical propertie sof the enhancement-mode H-diamond MOSFETs?
4. What is the reason for the low leakage current density for this thin SiOx film? Do you think that the AlOx still exists because of the gate electrode Al oxidization?
5. Please modify the equation (3) from “=” to “+”.
Author Response
Dear Reviewers,
First of all, we would like to express our great thanks to you for your carefully reading our manuscript and providing us with very helpful suggestions. We appreciate your comments very much.
We have carefully revised the overall manuscript based on your comments and suggestions. The revised part has been marked using red font in the revised manuscript.
Best regards
We would like to resubmit the revised manuscript to Materials. Thank you for your consideration and time.
Sincerely yours
Shi He
Hong-Xing Wang
Reviewer 2 Report
The present contribution seems to be the natural continuation of a previous study published in Appl. Phys. Lett. (S. He et al, Appl. Phys. Lett. 120, 132102 (2022); https://doi.org/10.1063/5.0085935). The MOSFET presented has not outstanding properties, in particular Id and µ are relatively low. However, it normally-off behaviour probably caused by unoxidized Sn at the diamond interface is a very attracting aspect. This aspect can be investigated using either in-situ of the XPS Ar-milling or angle resolved XPS to evidence the variation of stoichiometry along the 5nm SnOxthickness, see for example respectively:(i)G. Alba et al, Nanomaterials 2020, 10, 1193; doi:10.3390/nano10061193.
The paper can be published after introduction of minor modifications. We ask to modify the following points:
1.- Page 1, line 37, the conductivity generated by the H-treatment is due to a “Surface transfert doping” process. This should be mentioned with a corresponding reference. We suggest: K.G. Crawford et al, Progress in Surface Science 96, 100613 (2021), “Surface transfert doping of diamond: A review”.
2.- Page 2, line 69: How much time has been exposed the sample after the H-plasma? This should be explained.
3.- Page 2, line 74: The oxidation treatment at 100ºC is carried out in the air? This should be also explained.
4.- Page 3, Fig.2b caption, please change it to: “XPS spectrum at the Sn 3d photoelectron energy range that evidence the partial thermal oxidation of Sn.”. As mentioned above, the demonstration that the Sn is not oxidized at the Sn/diamond interface can by obtained by angle resolved or by Ar-milling in-situ of the XPS.
5.- Page 3, Fig.1c, the gate oxide is not SnO2but SnOx. Indeed, for SnO2the MOSFET should be normally – ON. The author suspects that the unoxidized Sn at the diamond interface induce a Schottky behaviour. The referee agrees with this explanation. But, moreover, the gate contact (aluminium) did not cover the whole SnOxand thus to allow to open the channel, the Sn at the diamond interface should be metallic to uniform the surface potential. Thus both things should be introduced: (i) change the SnO2in SnOxon the figure, (ii) give an explanation for the channel opening mechanism: metallic Sn is required at the interface.
6.- Page 5, figure 3a, from my point of view, the leakage is relatively high, however, in the text, it is defined as “small” (line 137) and “low leakage…” in the conclusion. Can the author compare it with some reference from literature to motivate such small and low qualification of leakage?
7.- From my point of view, the density of carrier generated by the surface transfer doping should be relatively low (does it have some connection with the fixed charge density in the range of 1011cm-2which is more than one order of magnitude lower than values usually reported after H-treatment?) which explains the relative low value of the drain current (1A/mm is straightforward, see for example K. Hirama et al, Jpn J. of Appl. Phys. 51, 901121 (2012)). The time in the air after the H-treatment and before the Sn deposition can be an important point. Can the authors introduce in the discussion some comments?
Author Response

(The authors gave the same response as above.)

Reviewer 3 Report
In this work authors report an investigation on the H-diamond FET with SnOx dielectric layer deposited by thermal oxidation of Sn. The authors describe the technique of producing H-diamond FET very well, and carry out detailed research. The topic is not original or relevant to the field, but it addresses a specific gap in the field. Compared with other published subjects this one has more detailed research of the selected materials
The article can be accepted after minor correction.
Line 17 - Expand the FET abbreviation. Not everyone knows what it is.
Line 87 - "FWHM ... 0.0105" -no unit - How was this quantity determined - what function was used?
Line 122 - " ... 100 k Hz, 500 k Hz, 1M Hz, 5M Hz" - kHz, MHz - We write the prefixes together with the unit.
Line 163 - What is the uncertainty of the measurement points in Figure 6?
Conclusion - The conclusions do not contain any relevant information other than the summary of the results. Need for a more detailed description and potential application.
Author Response

(The authors gave the same response as above.)
